



# COVID-19 lockdowns highlight a risk of increasing ozone pollution in European urban areas

Stuart K. Grange[1,2], James D. Lee[2], Will S. Drysdale[2], Alastair C. Lewis[2,3], Christoph Hueglin[1], Lukas Emmenegger[1], and David C. Carslaw[2,4]

[1]Empa, Swiss Federal Laboratories for Materials Science and Technology, Überlandstrasse 129, 8600 Dübendorf, Switzerland
[2]Wolfson Atmospheric Chemistry Laboratories, University of York, York, YO10 5DD, United Kingdom
[3]National Centre for Atmospheric Science, University of York, Heslington, York, YO10 5DD, United Kingdom
[4]Ricardo Energy & Environment, Harwell, Oxfordshire, OX11 0QR, United Kingdom

**Correspondence:** Stuart K. Grange (stuart.grange@empa.ch); David C. Carslaw (david.carslaw@york.ac.uk)

**Abstract.**

In March 2020, non-pharmaceutical interventions in the form of lockdowns were applied across Europe to urgently reduce the transmission of SARS-CoV-2, the virus which causes the COVID-19 disease. The near-complete shutdown of the European economy had widespread impacts on atmospheric composition, particularly for nitrogen dioxide ($NO_2$) and ozone ($O_3$). To investigate these changes, we analyze data from 246 ambient air pollution monitoring sites in 102 urban areas and 34 countries in Europe between February and July, 2020. Counterfactual, business as usual air quality time series are created using machine learning models to account for natural weather variability. Across Europe, we estimate that $NO_2$ concentrations were 34 and 32 % lower than expected for traffic and urban-background locations while $O_3$ was 30 and 21 % higher (in the same environments) at the point of maximum restriction on mobility. The European urban $NO_2$ experienced in the 2020 lockdown was equivalent to that which might be anticipated in 2028 based on average trends since 2010. Despite $NO_2$ concentrations decreasing by approximately a third, total oxidant ($O_x$) changed little, suggesting that the reductions of $NO_2$ were substituted by increases in $O_3$. The lockdown period demonstrated that the expected future reductions in $NO_2$ in European urban areas are likely to lead to a widespread increase in urban $O_3$ pollution unless additional mitigation measures are introduced.

## 1 Introduction

On December 31, 2019, a cluster of unexplained pneumonia cases in Wuhan, Hubei, China was reported to the World Health Organization (WHO) (World Health Organization (WHO), 2020a; Wu et al., 2020). Subsequent research in January, 2020 identified the disease to be caused by a previously unknown betacoronavirus (SARS-CoV-2), and the disease was given the name coronavirus disease 2019 (COVID-19) (Zhou et al., 2020; World Health Organization (WHO), 2020c). Due to rapid human-to-human transmission and the introduction of the virus to countries outside China, cases of COVID-19 were soon detected in all continents of the world, with the exception of Antarctica, and on March 11, the WHO declared a COVID-19 pandemic (World Health Organization (WHO), 2020b).





Europe was named the epicentre of the pandemic on March 13, and most European countries undertook unprecedented non-pharmaceutical interventions to reduce the transmission rate of SARS-CoV-2 in early or mid-March (BBC, 2020; Dehning et al., 2020; Remuzzi and Remuzzi, 2020). The exact nature and duration of the measures varied by country, but collectively

25    they are often referred to as "lockdowns" (Ruktanonchai et al., 2020). The lockdowns generally resulted in the closure of all shops, schools, universities, and restaurants with the exception of supermarkets, pharmacies, and other services deemed essential. Working from home whenever possible was encouraged and some countries also controlled, or restricted travel, exercise, and leisure activities. All these measures created a situation where European economic activity was reduced to a bare minimum within a matter of days, and mobility of the European population was severely altered (Figure 1).

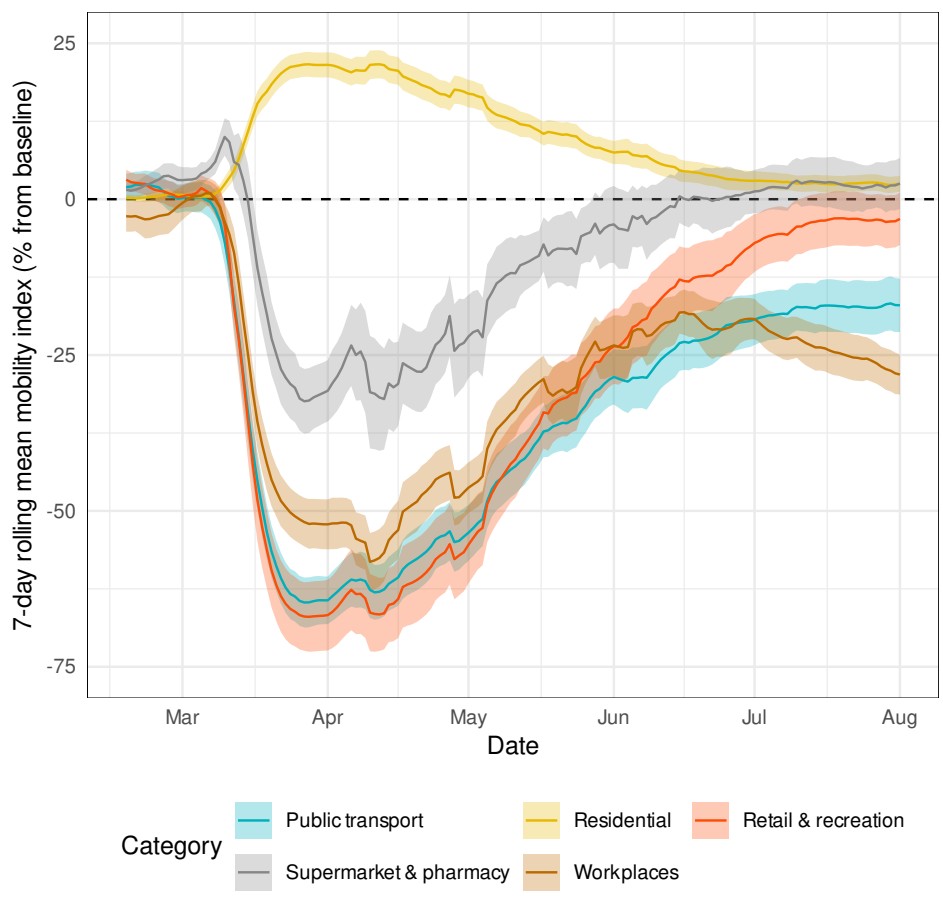

**Figure 1.** European mobility changes based on © Google's mobility indices between February and July, 2020 (© Google, 2020).

30    The rapid reduction of economic activity had many positive environmental impacts with the improvement of air quality being widely reported, especially via striking satellite observations of column $NO_2$ (Liu et al., 2020; Patel et al., 2020; Venter et al., 2020). Reductions of $CO_2$ emissions have also been reported globally due to heavily curtailed economic activities (Le





Quéré et al., 2020; Forster et al., 2020). Many of the reports of improved air quality were preliminary, and further research was required to fully understand and quantify the improvements observed throughout Europe, particularly after accounting for meteorological factors (Grange et al., 2020; Carslaw, 2020; Lee et al., 2020; Wang et al., 2020).

The European lockdowns can be thought of and approached as an air quality 'experiment' where economic activity was curtailed to near-minimum levels. Questions can be asked from the data such as: what were the results, how do they compare to other planned interventions such as low emission or clean air zones, and whether the observations were inline with what would be expected? The rate and severity of the changes imposed on European populations due to the lockdowns is something that previously could only be investigated by atmospheric modeling. Therefore, the COVID-19 lockdowns have provided a unique 'real-world modeling scenario' which represents a plausible future with far fewer internal combustion engine vehicles in use across Europe.

Here, we report an analysis based on counterfactual business as usual scenarios using predictive machine learning models. This allows for robust comparisons of the observed concentrations of air pollutants with those which would have been expected without the lockdown measures. The primary objective of this study is to report the response of $NO_2$ and $O_3$ concentrations throughout European urban areas caused by mobility restrictions due to COVID-19 lockdown measures. A secondary objective is to outline the implications for European air quality management which the dramatic changes in population mobility exposed.

## 2 Materials and methods

### 2.1 Data

Up-to-date (UTD) hourly $NO_2$ and $O_3$ motioning data were retrieved from the European Air Quality Portal (European Environment Agency, 2019) for the period between 2018 and 2020 for 102 urban areas in 33 European countries (Figure 2). For the 34th country, the United Kingdom, observations were directly retrieved from the countries' individual (England, Wales, and Scotland) and national networks (Automatic Urban and Rural Network; AURN) (Department for Environment Food & Rural Affairs, 2020).

The 102 urban areas were chosen because they are the capital, a "principal", or a particularly relevant city for the included European countries (Figure 2). In each urban area, at least one representative traffic site and at least one urban-background site were chosen (if available) to represent the area. Notably, UTD data are not validated, are subject to change, and will only be finalised (at the time of writing) in 11 months time (the deadline is September, 2021). However, the time series were screened for undesirable features such as calibration issues, frequent missing data, or long periods of no reported data. Time series with such obvious issues were not included in the analysis. Unfortunately, oxides of nitrogen ($NO_x = NO_2 + NO$) data were not available because most countries which participate in the UTD process do not report $NO_x$ (or NO) since it is not a regulated, ambient pollutant in Europe (Grange, 2019). Additionally, total oxidant ($O_x = NO_2 + O_3$) was calculated (in ppb) and included in the analysis as a third variable.





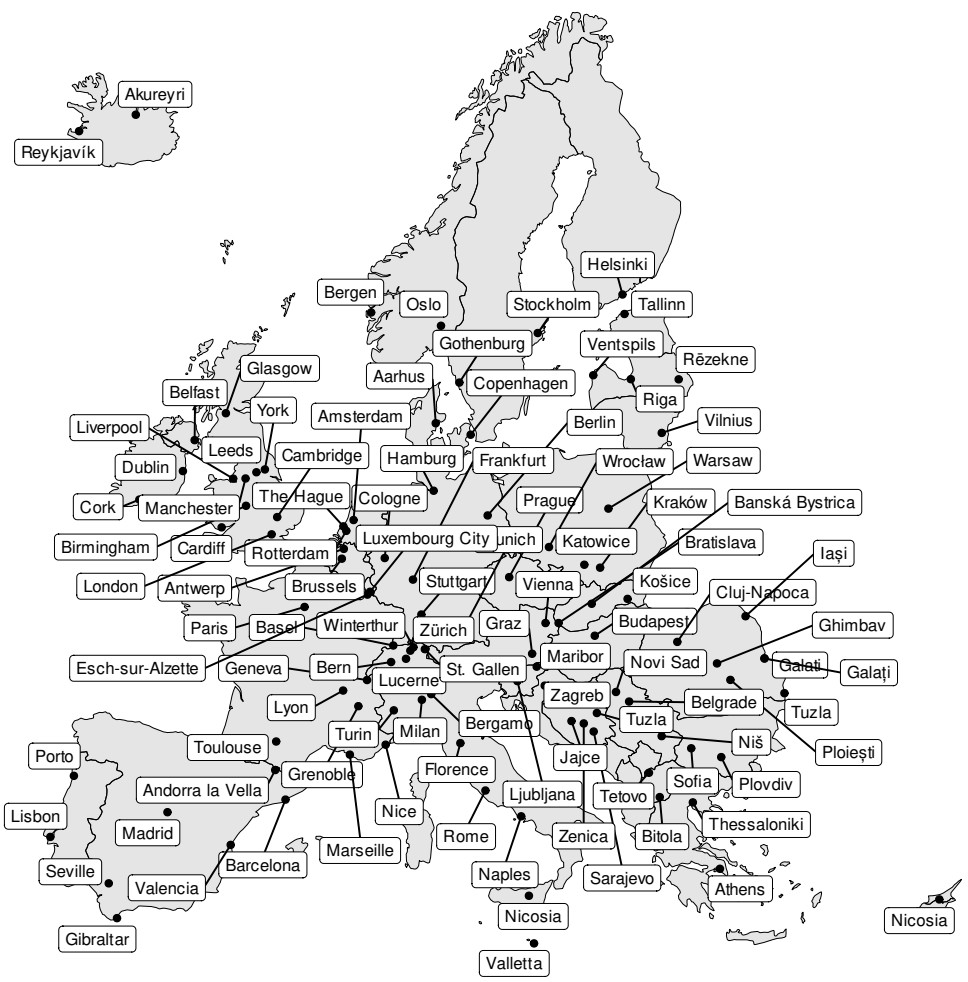

© OpenStreetMap contributors. Distributed under a Creative Commons BY-SA License

**Figure 2.** The 102 European urban areas included in the data analysis.

Hourly surface-based meteorological data were downloaded from the Integrated Surface Database (ISD). For the 102 urban
areas, these sites were generally airports (NOAA, 2016; Grange, 2020). A total of 246 air quality monitoring sites and 91
meteorological sites were included in the analysis. For details of the sites, see the tables available online[3].

In the current work, we focus on changes in the concentrations of $NO_2$ and $O_3$ at urban-traffic and urban-background loca-
tions. $NO_2$ and $O_3$ in such locations are strongly influenced by local road vehicle emissions and not, for example, transboundary
contributions, which would be the case for particulate matter ($PM_{2.5}$ and $PM_{10}$). Furthermore, the concentrations of $NO_2$ and
$O_3$ in urban areas are strongly influenced by local meteorological effects. Generally, traffic sites are located in close proximity
to roads, and pollutant concentrations are forced by local vehicular emissions. The urban-background classification is more

---

[3]Temporary location: http://skgrange.github.io/www/data/data_analysis_links.html





varied, but can be thought as environments away from the immediate vicinity of roads and industrial facilities but are still located within an urban area.

## 2.2 Business as usual (BAU) modeling

A central issue when considering changes in atmospheric concentrations due to an intervention is whether the change is due to variations in meteorological conditions or emission source strength (Grange and Carslaw, 2019). This problem is widespread and affects time scales from hours to years. It is particularly important in 'before-after' studies where meteorological change, rather than changes in emission source strength, can easily dominate the variation in concentrations. This ambiguity can be somewhat reduced by averaging over several years to account for past inter-annual variability. However, this approach cannot
account for the significant impact that meteorology may have on a specific observation period.

In the current context of the changes in activities brought about by COVID-19 lockdowns, the changes are over a duration of several months and span a period from spring to summertime conditions. This period straddles important natural changes in meteorological conditions and atmospheric composition. For example, during February, 2020 the UK and much of western Europe experienced exceptionally high mean wind speeds due to storms Ciara, Dennis, and Jorge. Surface wind speed
records in Southern England suggest February, 2020 had the highest mean wind speed of any month for over 40 years. This demonstrates that the state of the atmospheric dispersion across Europe at the time of COVID-19 lockdowns was different than experienced in previous years. Similarly, urban-background concentrations of $O_3$ in the northern hemisphere tend to increase from the beginning of the year and peak in April, which will also influence $NO_2$ (Monks, 2000). These, and other factors suggest that considerable care is needed for the quantification of an intervention such as the COVID-19 lockdowns on surface
concentrations of primary and secondary pollutants.

To address the above issues, random forest models were trained to explain hourly mean $NO_2$ and $O_3$ concentrations using surface meteorological and time explanatory variables for each monitoring site (Breiman, 2001). The explanatory variables used were: wind direction, wind speed, air temperature, relative humidity, atmospheric pressure (if available in the ISD database), a trend term in the form of Unix date, a seasonal term in the form of Julian day, weekday, and hour of day. The following random
forest hyper-parameters were kept constant for all models: 300 trees, three variables to split at each node, and a minimal node size of five. The training period spanned just over two years and was between January 1, 2018 and February 14, 2020. The training-testing split percentage was 80 and 20 respectively. From February 14 to July 31, 2020, the models were used in predictive mode to predict pollutant concentrations based on the observed meteorological variables.

The models' predictions can be thought of as business as usual (BAU) scenarios based on past behaviour of pollutant
concentrations and the weather which was experienced after February 14 at each monitoring site. Thus, the model represents a *counterfactual* which observed concentrations can be compared with (for example, see Figure 3).

February 14 to March 1, 2020 was considered a validation period where the models' skill were checked for adequate performance. Summaries of the models' performance based on the random forest model objects and predictions during the validation period in the form of $R^2$ are shown in Figure A1. From the start date of the lockdowns (the earliest was March 9 in Italy), the
application period began and gave estimates of BAU, *i.e.*, what concentrations would have been if the lockdown measures were





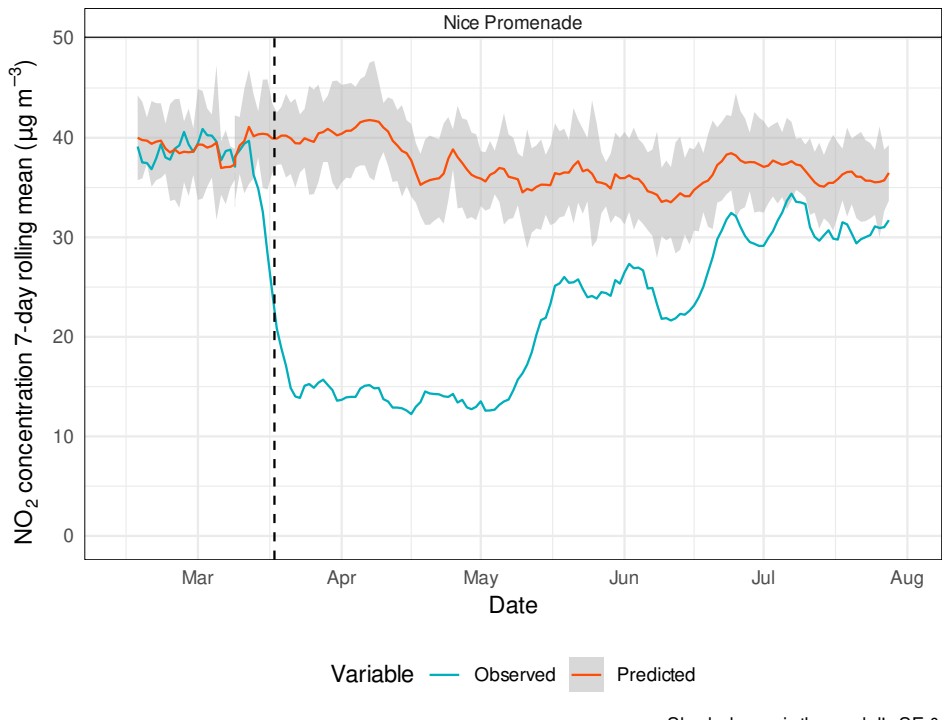

**Figure 3.** A $NO_2$ example where the observed concentrations clearly diverged from the business as usual (BAU) scenario for the Nice Promenade (France) traffic monitoring site between February and July, 2020.

not implemented. The modeling was conducted using the **rmweather** R package (Grange et al., 2018; Grange and Carslaw, 2019; Grange, 2018).

During the validation phase, a number of models showed bias in prediction, most notably, $NO_2$ was under-predicted at many locations. The under-prediction was on average, -3.7 $\mu g\,m^{-3}$ (95 % CI [-4.2, -3.3]). This under-prediction was most likely caused by already-curtailed economic activity and reduced emissions throughout Europe at the very end of February and the beginning of March, *i.e.*, before the formal lockdowns were implemented. The beginning of 2020 was also mild in respect to ambient temperature and rather windy at most locations (discussed above) which may have resulted in some models under-predicting concentrations at this time of the year. For consistency and to create a reference point in time, the model predictions were corrected by calculating the model offset validation phase (February 14 to March 1) and subtracting this offset from the predictions. This ensured that the counterfactual predictions were calibrated at the start of the application phase and represented the changes in concentrations after March 1, 2020.



### 2.2.1  Change point analysis

To link $NO_2$ and $O_3$ concentration changes in March–April, 2020 to the lockdown restrictions placed on European populations, change point models were calculated. These change point models were conceptually simple – an intercept change was the

expected *a priori* assumption. There were two motivations for these change point models. The first was to identify both the time, and magnitude of concentration response with an objective, data-driven approach rather than using a subjective and manual classifier. The second was to use such a technique to identify an atmospheric response after an intervention (an unplanned one in this case) which is a general goal of air quality data analysis.

The change point logic was implemented with the **mcp** R package with Bayesian inference (Lindeløv, 2020). To detect

the change points, three Markov chains were run with 9000 iterations. The change point models tested the delta between the observed and counterfactual, however, the change-points were calibrated back to their pre-lockdown concentrations to conduct the (relative) percentage change calculations.

### 2.2.2  Presentation of results

When presenting the results of the analysis, most time series are displayed as seven-day rolling means. These rolling means act

as a smoothing filter to make patterns clearer and remove the day-to-day variations generally seen in air quality monitoring data. Thirty-four countries were included in the analysis (Figure 2), but to avoid overwhelming plots and figures, a consistent set of six European countries (France, Germany, Italy, Spain, Switzerland, and the United Kingdom) were chosen to be displayed when discussing the counties' air quality patterns.

## 3  Results and discussion

### 3.1  Mean concentration changes

For all 34 European countries analysed, the observed concentrations of $NO_2$ were lower than those predicted by the counterfactual business as usual (BAU) scenarios between February 14 and July 31, 2020 (deltas ($\Delta$) between the observed concentrations and predicted counterfactual shown in Figure 4). The reductions of $NO_2$ were greater in both an absolute and relative sense at the sites classified as either roadside or traffic environments compared to urban-background locations which can be explained

by $NO_2$ being primarily a traffic-sourced pollutant (Grange et al., 2017). The impacts of vehicle-flow reductions during the lockdowns were more dramatic in the close proximity of roads when compared to more distant urban-background locations.

Mean $O_3$ concentrations increased at a similar magnitude to which $NO_2$ decreased throughout Europe between February and July, 2020 (Figure 4). Like $NO_2$, $O_3$ at roadside locations showed a greater divergence from the BAU predictions than urban-background sites. The near-mirror image of $NO_2$ and $O_3$ can be explained by the relationship between $NO_x$ and $O_3$.

The reduction of $NO_x$ emissions and concentrations across Europe drove decreased $O_3$ destruction via the NO titration cycle during this period. In many countries, the 8-hour legal limit for $O_3$ of $120\,\mu g\,m^{-3}\,8\,h^{-1}$ was breached during this time period.



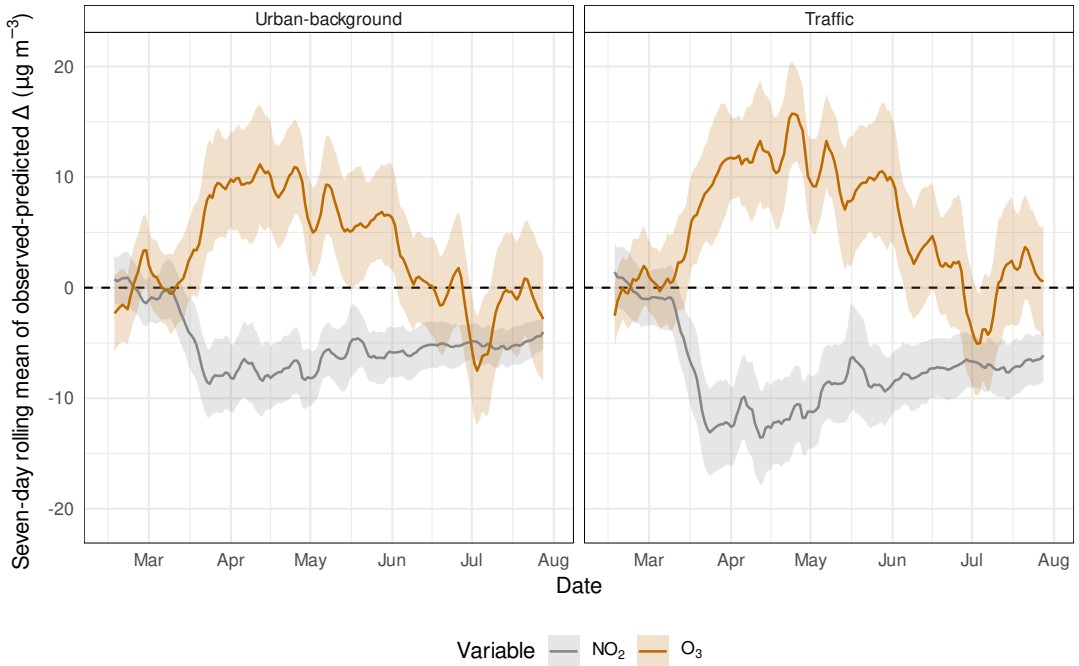

**Figure 4.** Seven-day rolling means of the observed-predicted concentrations deltas for NO$_2$ and O$_3$ for all European sites analysed between February 14 and July 31, 2020.

Unlike NO$_2$ where concentrations remained below their BAU estimates until the end of the analysis period, O$_3$ concentrations returned to their expected values by the end of July, 2020.

### 3.2 Timing of changes

Figure 4 clearly indicates that concentrations in the first half of 2020 diverged from what was predicted by the counterfactual modelling. To objectively identify the date and magnitude of maximum divergence, change points were identified with a data-driven approach using Bayesian inference. The mean dates when NO$_2$ started to diverge at their greatest extent from the BAU scenarios along with national lockdown dates for six European countries are displayed in Figure 5. For the complete set of dates for all countries included in the analysis, see Table A1.

For NO$_2$, the change points were between seven days before and seven days after the countries' lockdown date (excluding the outlier of Denmark). For O$_3$, this range was greater, between -12 and 8 days. Italy was the first country in Figure 5 where change points were identified for NO$_2$ concentrations on March 13, 2020 and this was four days before Italy's nationwide lockdown date while Spain's NO$_2$ change point was the same as the country's lockdown date. Change points were often identified a day or two earlier than the lockdown date when the lockdown began on a Sunday or a Monday, for example, in

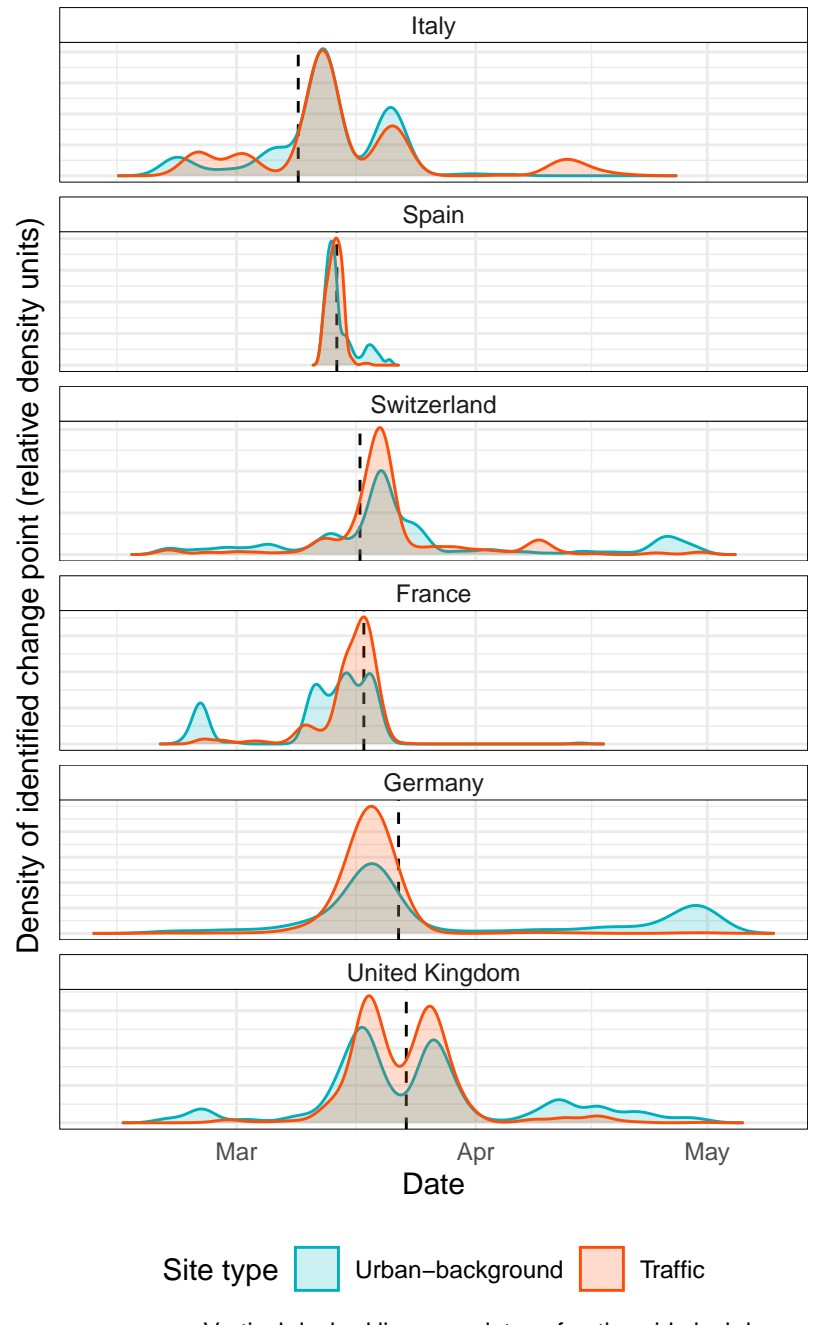

**Figure 5.** Estimated timing of changes to $NO_2$ concentrations for six European countries between March and May, 2020. The distribution shown for each country is the dimensionless probability distribution of the estimated change-point in concentration. The country panels are ordered by nationwide lockdown date.





Germany. For almost every site included in the analysis, the change points for $NO_2$ were ones of decreases while those for $O_3$ were increases (as seen in Figure 4).

    Figure 5 shows that some countries had very consistent changes in concentrations for the sites which were analysed, for example Spain. Changes in other counties were less consistent which may indicate regional differences within countries. The UK showed two peaks in density for the $NO_2$ change points which were separated by a week. This feature represents a two-

phase reduction in emissions because staggered lockdown measures were announced – the first was a set of recommendations for social distancing and not visiting restaurants and other social establishments (on March 16), while the second announcement (March 23) was one of a more strict lockdown.

    Although the identified change point dates for $NO_2$ were broadly consistent with the various countries' lockdown dates, the change points for $O_3$ were not aligned as closely (Table A1). There was also no correlation between the magnitude of

$NO_2$ reduction and the time required for an $O_3$ change point to be identified. This suggests that $O_3$'s secondary generation processes did not immediately respond to reductions of ambient $NO_x$ concentrations after lockdowns were imposed due to less NO titration. For this process to be identifiable, $O_3$ generation must occur, and this requires sunlight. Therefore, the lack of sunny conditions in some urban areas around the time of the $NO_2$ atmospheric response may have resulted in varying duration lags before changes in $O_3$ could be observed.

**3.3   Concentration changes among different countries**

  At a European level, maximum divergence of $NO_2$ and $O_3$ from the counterfactual predictions was reached in late-March, 2020 (Figure 4). However, there was some diversity among European country $NO_2$ and $O_3$ divergence from their counterfactuals for the analysis periods (Figure 6). All countries analysed passed their maximum divergences for $NO_2$ and $O_3$ in late-April, and the shape of the recovery is of a "swoosh" with a sharp plunge away from the counterfactual around the date of the lockdown

implementations (Figure 6), but the rapid plunge was followed by a slower, and more gradual return to the BAU until the end of July. This pattern is very much reminiscent of the mobility changes shown in Figure 1.

    Some countries experienced a smaller reduction in $NO_2$ than others. Germany and Switzerland for example, experienced lower $NO_2$ reductions when compared to France, Italy, and Spain. Some countries' greater reductions in ambient $NO_2$ concentrations could be explained by the level of "stringency" of the countries' lockdowns and resulting changes in mobility

(Hale et al., 2020; © Google, 2020). For example, Germany and Switzerland's measures were very strong recommendations with few legally enforceable restrictions on recreational or leisure activities, while France, Italy, and Spain had more stringent requirements where movement and travel were restricted and enforced in a much stronger manner. It is very likely that these different levels (or enforcement) of restrictions had implications for emissions of atmospheric pollutants. However, meteorological conditions, perhaps similar synoptic scale patterns likely played a role in the differences observed among the countries

too.

    After late-April, concentrations moved towards their predicted counterfactual values and this continued to the end of the analysis period (Figure 6). Some European countries began to remove lockdown restrictions in the second half of April which increased traffic-sourced emissions, and this is consistent with the observations in Figure 4 and Figure 6. $O_3$ concentrations



**Figure 6.** Seven-day rolling means of the observed-predicted concentrations deltas for $NO_2$, $O_3$, and $O_x$ for six selected countries in Europe between February 14 and July 31, 2020.

returned to approximately their BAU levels by the end of July, but $NO_2$ had yet to do so at the end of the analysis period,
with the exception of Italy. This indicates that $NO_x$ emissions (mostly traffic-sourced) had not yet reached their estimated BAU
levels by the end of July across most of Europe after the country lockdowns were released.



## 3.4 Quantifying the changes in concentrations

The change point dates identified by Bayesian inference shown in Figure 5 and Table A1 were used to classify the time series as pre-lockdown, within lockdown, or post-lockdown periods. With this classification, concentrations were compared to calculate concentration deltas and percentage changes. At a European level, the mean $NO_2$ percentage changes for $NO_2$ at traffic and urban-background sites were -34 % (95 % CI [-36, -31]) and -32 % (95 % CI [-35, -29]) respectively (which equalled concentration reductions of -11 and -7 $\mu g\,m^{-3}$). The European annual $NO_2$ standard is $40\,\mu g\,m^{-3}\,y^{-1}$, and the mean reduction of $11\,\mu g\,m^{-3}$ is 27 % of the legal limit (European Commission, 2019). For $O_3$, the mean European percentage change for traffic and urban-background sites were estimated at 30 % (95 % CI [26, 35]) and 21 % (95 % CI [18, 24]), and the concentration changes were 12 and $9\,\mu g\,m^{-3}$ respectively. The concentration deltas and percentage changes attributed to the European lockdown measures are listed by country and site type in Table 1.

To put these concentration changes into context, $NO_2$ and $O_3$ trend analysis between 2010 and 2019 for the 246 sites was conducted. Based on the sites which had a complete data record, the mean trends were -1.44 and -0.72 $\mu g\,m^{-3}\,y^{-1}$ for $NO_2$ at traffic and urban-background locations, while $O_3$ trends in the same environments were 0.2 and $0.49\,\mu g\,m^{-3}\,y^{-1}$. Therefore, at the roadside, the mean reduction of $NO_2$ across Europe due to the COVID-19 lockdown measures was equivalent to that of 7.6 years of continued concentration reduction, or equivalent to the anticipated European atmosphere in 2028 (Figure 7). $O_3$ however, increased at an equivalent of 17 years of the rate of change determined by trend analysis in urban-background locations.

The changes at traffic sites will strongly reflect the influence of changes in traffic activity in close proximity to each site for $NO_x$, $NO_2$ and $O_3$. Close to roads, the origins of $NO_2$ can be thought of as the combination of a background component, a component which is generated from the fast reaction between vehicular NO emissions and $O_3$, and directly emitted (primary) $NO_2$. The primary $NO_2$ contribution is known to have decreased in recent years from a peak around 2010. In London for example, the analysis of 35 traffic-influenced sites showed a reduction in the mean $NO_2/NO_x$ vehicle emission ratio from around 25% in 2010 to about 15% in 2014, (Carslaw et al., 2016) while at a European level, the $NO_2/NO_x$ emission ratio peaked at 16 % (also in 2010) (Grange et al., 2017). This decrease is believed to be driven by improvements in selective catalytic reduction control systems used on vehicles to reduce $NO_x$ and also to the effect of ageing of diesel oxidation catalysts (Carslaw et al., 2019).

The decrease in primary $NO_2$ emissions over the past decade would have acted to reduce ambient $NO_2$ concentrations close to roads. Indeed, if the traffic reductions experienced across Europe through country-wide lockdowns had occurred closer to 2010, the reductions in road vehicle $NO_2$ emissions would have been much more important in affecting ambient concentrations than was experienced in early 2020.

The posterior draws (a type of model prediction) from the change point models show that in some countries, the reduction of traffic volumes during the COVID-19 lockdowns reduced $NO_2$ concentrations to those which are experienced at urban-background locations (United Kingdom shown in Figure 8). The roadside increment or enhancement of $NO_2$ was essentially eliminated by reducing traffic to the levels which were experienced while in the lockdown state. However, as discussed above





**Table 1.** Mean concentration deltas/differences and percentage changes of $NO_2$, $O_3$, and $O_x$ for different countries and site types attributed to COVID-19 lockdown measures in March, 2020. Values which are missing indicates that there were not data and NC indicate no change point was identified.

| Country | Site type | $NO_2$ $\Delta$ ($\mu g\,m^{-3}$) | % change | $O_3$ $\Delta$ ($\mu g\,m^{-3}$) | % change | $O_x$ $\Delta$ (ppb) | % change |
|---|---|---|---|---|---|---|---|
| Andorra | Traffic | – | – | – | – | – | – |
| Andorra | Urban-back. | -19.8 | -59.7 | 16.1 | 43.0 | -3.4 | -9.8 |
| Austria | Traffic | -7.6 | -24.5 | – | – | – | – |
| Austria | Urban-back. | -5.2 | -23.1 | 11.3 | 19.5 | 4.3 | 11.2 |
| Belgium | Traffic | -10.8 | -45.3 | 5.0 | 10.5 | -2.2 | -6.5 |
| Belgium | Urban-back. | -9.5 | -38.4 | 8.9 | 19.2 | 2.4 | 6.5 |
| Bosnia and Herzegovina | Traffic | – | – | – | – | – | – |
| Bosnia and Herzegovina | Urban-back. | -1.8 | -11.9 | 1.4 | 15.0 | -1.3 | -3.4 |
| Bulgaria | Traffic | -13.8 | -29.5 | 14.0 | 29.6 | 0.9 | 2.2 |
| Bulgaria | Urban-back. | -10.4 | -34.2 | 13.9 | 33.6 | 3.0 | 8.4 |
| Croatia | Traffic | -16.2 | -42.3 | – | – | – | – |
| Croatia | Urban-back. | -12.4 | -43.9 | 21.5 | 34.1 | 4.4 | 9.6 |
| Cyprus | Traffic | -15.3 | -47.0 | – | – | -2.8 | -7.2 |
| Cyprus | Urban-back. | -16.7 | -59.7 | 6.1 | 10.9 | -5.0 | -11.8 |
| Czechia | Traffic | NC | NC | – | – | – | – |
| Czechia | Urban-back. | NC | NC | – | – | – | – |
| Czechia | Urban-back. | – | – | 9.0 | 18.3 | 4.9 | 13.8 |
| Denmark | Traffic | -6.7 | -28.0 | 15.7 | 31.7 | 3.9 | 9.8 |
| Denmark | Urban-back. | -4.2 | -49.0 | 7.6 | 12.3 | 3.1 | 8.4 |
| Estonia | Traffic | -5.0 | -35.2 | 0.7 | 1.3 | -1.8 | -5.2 |
| Estonia | Urban-back. | -2.4 | -29.2 | 6.4 | 10.7 | -0.4 | -1.2 |
| Finland | Traffic | -9.4 | -42.5 | – | – | – | – |
| Finland | Urban-back. | -4.3 | -34.1 | – | – | – | – |
| France | Traffic | -20.3 | -54.2 | – | – | – | – |
| France | Urban-back. | -11.2 | -44.1 | 13.9 | 35.0 | -4.9 | -12.1 |
| Germany | Traffic | -10.5 | -29.3 | 15.1 | 37.3 | 3.0 | 7.5 |
| Germany | Urban-back. | -4.9 | -21.6 | 8.8 | 16.6 | 3.5 | 9.1 |
| Greece | Traffic | -12.3 | -37.1 | – | – | -1.1 | -0.4 |
| Greece | Traffic | – | – | NC | NC | – | – |
| Greece | Urban-back. | -9.5 | -43.9 | – | – | -3.8 | -8.5 |
| Greece | Urban-back. | – | – | NC | NC | – | – |
| Hungary | Traffic | NC | NC | – | – | – | – |
| Hungary | Urban-back. | NC | NC | – | – | – | – |
| Hungary | Urban-back. | – | – | 5.0 | 15.7 | -4.2 | -11.4 |
| Iceland | Traffic | -5.3 | -33.7 | – | – | – | – |
| Iceland | Urban-back. | -3.4 | -23.5 | – | – | – | – |
| Ireland | Traffic | – | – | – | – | – | – |
| Ireland | Urban-back. | -4.9 | -33.6 | – | – | -1.3 | -3.5 |
| Ireland | Urban-back. | – | – | NC | NC | – | – |
| Italy | Traffic | -17.3 | -31.9 | – | – | – | – |
| Italy | Urban-back. | -12.5 | -32.7 | 3.8 | 14.1 | -1.5 | -2.2 |
| Lithuania | Traffic | -7.0 | -25.9 | 13.8 | 34.3 | 2.8 | 7.3 |
| Lithuania | Urban-back. | -4.5 | -21.0 | – | – | – | – |
| Luxembourg | Traffic | -15.5 | -53.2 | – | – | – | – |
| Luxembourg | Urban-back. | -10.3 | -47.0 | 9.6 | 17.0 | -0.1 | -0.3 |

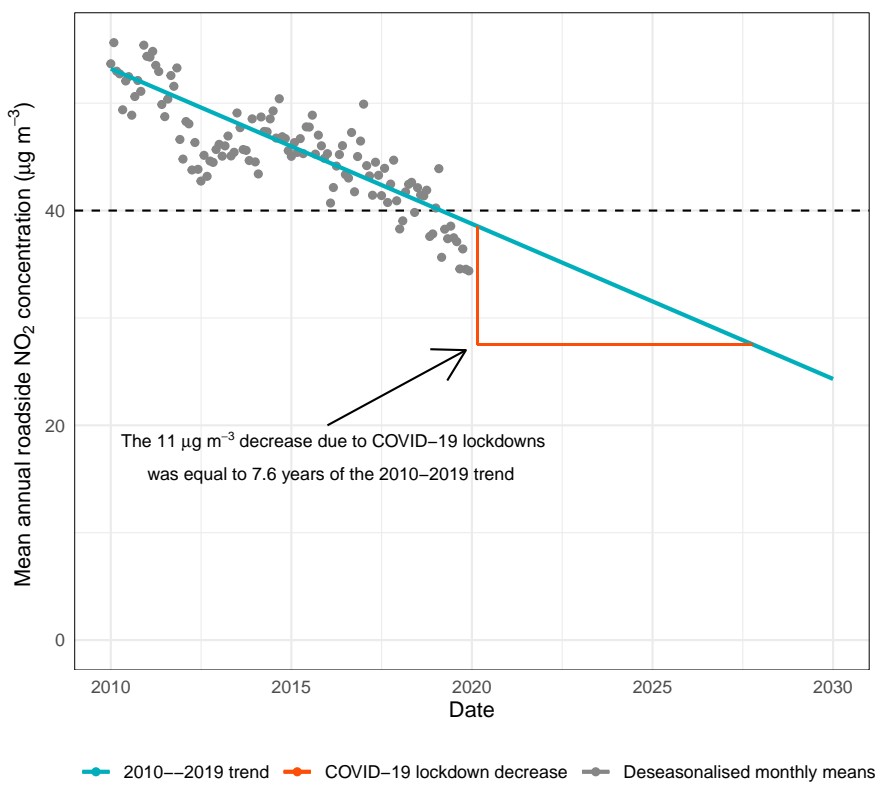

**Figure 7.** Mean European roadside $NO_2$ trend with the reduction of $NO_2$ concentrations attributed to the COVID-19 lockdowns put in context.

and shown in Figure 8, $O_3$ increased in response to the reductions of $NO_2$ and $O_x$ only altered very slightly. The same patterns in the United Kingdom were also experienced in other European countries such as France and Spain, but were not as clear for counties such as Switzerland and Germany.

## 3.5 $O_x$ – $NO_2$ and $O_3$ repartitioning

Figure 4 and Figure 6 demonstrate that $NO_2$ concentrations and emissions decreased throughout Europe due to the COVID-19 lockdown measures, especially at the roadside. However, the reduction of $NO_2$ was accompanied by an increase of $O_3$ at a similar magnitude and resulted in $O_x$ showing little change despite the large reductions in traffic-sourced $NO_2$ (for example, Figure 8).

Mean European changes in $O_x$ were variable between the two site environments. At traffic sites, $O_x$ decreased by -1 ppb
(-1.8 %; 95 % CI [-4, 0.7]) while in urban-background locations, $O_x$ increased by 0.7 ppb (2.1 %; 95 % CI [-0.2, 4]). In the case of the traffic sites, the modest decrease of $O_x$ can be partially explained by decreased emissions of primary $NO_2$ (Grange et al., 2017). However, in urban-background locations, $O_x$ remained nearly constant. This is a very important observation





**Table 1.** Table 1 continued.

| Country | Site type | NO$_2$ $\Delta$ ($\mu$g m$^{-3}$) | NO$_2$ % change | O$_3$ $\Delta$ ($\mu$g m$^{-3}$) | O$_3$ % change | O$_x$ $\Delta$ (ppb) | O$_x$ % change |
|---|---|---|---|---|---|---|---|
| Malta | Traffic | -13.2 | -38.7 | 10.0 | 15.4 | -4.1 | -8.1 |
| Malta | Urban-back. | – | – | – | – | – | – |
| Netherlands | Traffic | -6.2 | -28.3 | – | – | 1.3 | 3.5 |
| Netherlands | Traffic | – | – | NC | NC | – | – |
| Netherlands | Urban-back. | -3.5 | -21.2 | – | – | 4.1 | 11.2 |
| Netherlands | Urban-back. | – | – | NC | NC | – | – |
| North Macedonia | Traffic | -8.6 | -33.2 | – | – | -1.9 | -6.8 |
| North Macedonia | Traffic | – | – | NC | NC | – | – |
| North Macedonia | Urban-back. | – | – | – | – | – | – |
| Norway | Traffic | -7.7 | -30.0 | – | – | – | – |
| Norway | Urban-back. | -2.8 | -17.1 | – | – | 0.9 | 2.2 |
| Norway | Urban-back. | – | – | NC | NC | – | – |
| Poland | Traffic | -11.7 | -27.6 | – | – | – | – |
| Poland | Urban-back. | -3.6 | -12.7 | 7.1 | 15.1 | 2.1 | 5.5 |
| Portugal | Traffic | -25.9 | -53.8 | 20.2 | 46.8 | -10.7 | -24.6 |
| Portugal | Urban-back. | -11.9 | -40.5 | 13.8 | 26.8 | 4.7 | 12.1 |
| Romania | Traffic | -5.8 | -7.2 | – | – | – | – |
| Romania | Urban-back. | -7.5 | -26.3 | 13.0 | 39.9 | -0.5 | -0.5 |
| Serbia | Traffic | – | – | – | – | – | – |
| Serbia | Urban-back. | -10.4 | -56.4 | 15.6 | 44.9 | -4.1 | -12.6 |
| Slovakia | Traffic | -6.8 | -19.5 | – | – | – | – |
| Slovakia | Urban-back. | – | – | – | – | – | – |
| Slovenia | Traffic | -9.6 | -30.5 | – | – | – | – |
| Slovenia | Urban-back. | -5.0 | -18.9 | 20.9 | 55.7 | 8.2 | 26.1 |
| Spain | Traffic | -22.8 | -57.2 | 21.0 | 61.9 | -1.5 | -2.8 |
| Spain | Urban-back. | -16.4 | -55.7 | 15.9 | 37.5 | -2.2 | -5.4 |
| Sweden | Traffic | -4.9 | -17.0 | – | – | – | – |
| Sweden | Urban-back. | -1.5 | -12.5 | 6.5 | 12.2 | 0.6 | 2.0 |
| Switzerland | Traffic | -5.5 | -17.2 | 10.9 | 22.1 | 5.1 | 13.0 |
| Switzerland | Urban-back. | -3.3 | -10.1 | 11.7 | 21.7 | 5.2 | 14.4 |
| United Kingdom | Traffic | -14.4 | -50.8 | 14.4 | 45.8 | -3.8 | -8.3 |

for European air quality management. It suggests that the 34 % reduction of NO$_2$ concentrations was equalled by a similar absolute increase in O$_3$, which is clearly an undesirable outcome because of the deleterious effects of O$_3$ on population health,

buildings, and vegetation.

The repartitioning of NO$_2$ to O$_3$ is of importance from a public health perspective. As Williams et al. (2014) argue, there are good reasons from an atmospheric chemistry perspective to consider NO$_2$ and O$_3$ together in epidemiological studies, rather than either of the two pollutants separately in single-pollutant models. Indeed, Williams et al. (2014) found that there were larger associations (on mortality) for mean 24 hour concentrations of O$_x$ than for either O$_3$ or NO$_2$ individually. On this

basis, the current analysis suggest that the health impacts may have been small because O$_x$ concentrations changed little in urban environments. The analysis conducted here was exclusively concerned with daily mean O$_3$ concentrations, and does not

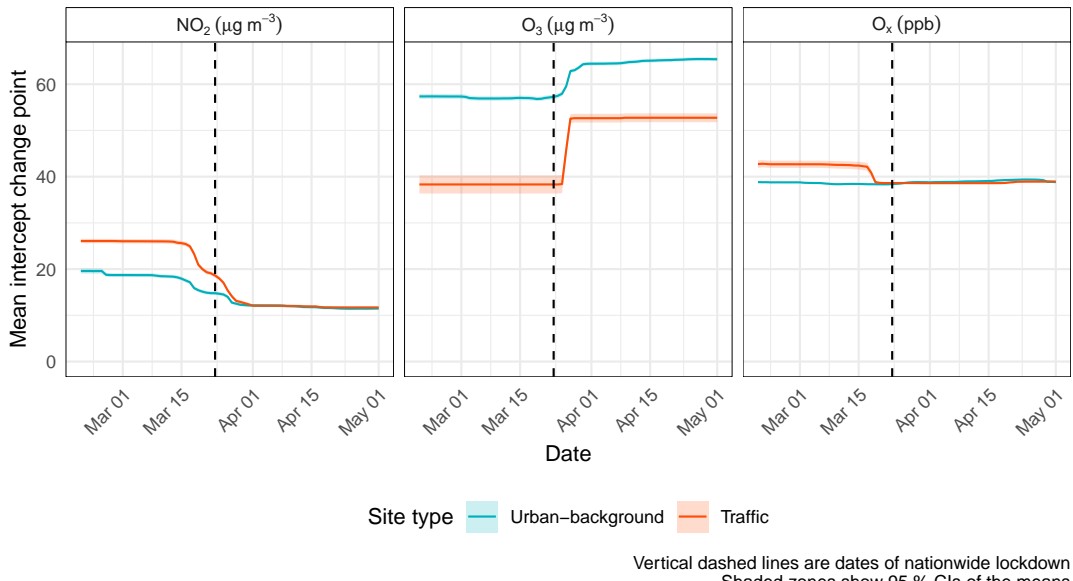

**Figure 8.** Posterior draws for $NO_2$, $O_3$, and $O_x$ two-intercept change point models for the United Kingdom between March and May, 2020.

explore the subtleties associated with peak and/or increases in daily minima $O_3$ concentrations which are also important when considering the deleterious effects of $O_3$.

Efficacious management of $O_3$ has proven to be a challenge in Europe and in many other locations around the world (Sillman,
1999; Wang et al., 2017; Chang et al., 2017; Li et al., 2019). The struggle with $O_3$ control is partly due to the highly non-linear chemistry of $O_3$ production based on the precursors volatile organic compounds (VOCs) and $NO_x$. There are two regimes: $NO_x$-sensitive and VOC-sensitive – and the $O_x$ analysis presented here strongly suggests that $O_3$ production is overwhelmingly VOC-sensitive across urban Europe. Therefore, if higher $O_3$ concentrations are to be avoided in the future where reductions in $NO_x$ emissions of the scale seen in lockdown are likely, enhanced control of VOC emissions will be critical in the European
urban environment. The prominence given to $NO_2$ as a pollutant following the dieselgate scandal of 2015 (Anenberg et al., 2017) has led to far more ambitious $NO_2$ emissions reductions policies in Europe than are currently in place for VOCs.

VOCs are only measured routinely in a few locations throughout Europe's urban areas, and represent a broad class of pollutant that are emitted from a wide range of sources. Whilst in the 1980s and 1990s VOC emissions were dominated by gasoline vehicle emissions (both tailpipe and evaporative) in more recent years their abundance has become increasingly
influenced by non-transport sources such as natural gas leakage and wider solvent use (Lewis et al., 2020).

Data from the London Eltham site, the only suburban VOC monitoring site in the UK, indicates that for many VOCs lockdown did not lead to significant changes in overall emissions or atmospheric concentrations (Figure A3). A conclusion from this albeit anecdotal evidence would be that further reductions in only traffic-related VOC emissions would not likely generate the desired air quality improvements in $O_3$ and that reducing emissions from other sectors would be essential.





Although out of scope for this current work, an obvious avenue for future research is to further explore how individual VOC concentrations responded during the lockdown periods in European urban areas in order to evaluate the proportion of VOCs that still come from traffic. This, combined with chemical modelling on a species by species basis to fully assess $O_3$ production chemistry, would help direct where future VOC reduction strategies should be focused. An analysis such as this would also strongly benefit from the access of $NO_x$ data which, arguably, would be a better pollutant to analyse than $NO_2$

from an emissions perspective. We strongly encourage the institutions which are involved with reporting ambient air quality data to the European Environment Agency to include $NO_x$ alongside the legally required $NO_2$ observations for the air quality community.

## 4    Conclusions

This work represents a classic air quality data analysis where atmospheric responses are linked to an intervention. In this case,

the intervention was an unplanned, likely unique, and extreme event with very different characteristics when compared to typical interventions such as the introduction of new emission standards and low emission zones. Despite the extreme nature of the COVID-19 lockdowns and their results being much more impactful on urban atmospheric composition than other policies over a short time period, the analysis still demonstrates the difficulty of detecting "change upon change" for atmospheric pollutants – especially for locations where concentrations are close to background. However, this analysis presents a robust

and portable framework for intervention analysis using a combination of machine learning-derived counterfactuals and change point analysis to identify the timing and magnitude of an atmospheric response.

Analysis of the effect of the European COVID-19 lockdowns on $NO_2$, $O_3$, and $O_x$ concentrations combining machine learning derived BAU modelling and Bayesian change point models indicate that $NO_2$ concentrations reduced by 34 % at roadside locations. However, the widespread reductions of $NO_2$ concentrations was accompanied by increases of $O_3$ at a

similar magnitude (30 %), and thus, $O_x$ altered only very slightly due to the lockdowns when considering Europe as a whole.

This insight has important implications for the implementation of future air quality management policies. The COVID-19 lockdown conditions give a glimpse of a realistic, and indeed likely, future environment where $NO_x$ emissions continue to reduce at their current rate, primarily because of the increasing stringency of vehicular emission standards (Carslaw et al., 2016; Grange et al., 2017). The future reduction of $NO_x$ concentrations will likely result in repartitioning of $O_x$ and the

increase of $O_3$ concentrations across most European urban areas. Although increases in European $O_3$ concentrations have been acknowledged, the further rise should be pre-empted by the European air quality management community through increased focus on VOC emission controls and the more holistic combined management of $NO_2$, $O_3$, and VOCs. This will allow for continued improvements to air quality in a general sense, rather than focusing on reductions of individual pollutants.



*Code and data availability.* The data sources used in this work are described and are publicly accessible in a temporary data repository

(http://skgrange.github.io/www/data/data_analysis_links.html). These data will be moved to a persistent data repository once the analysis is has been reviewed and finalised. Additional information about these data are available from the authors on reasonable request.

*Author contributions.* SKG and DCC conceived the research questions, conducted the analysis, and wrote the manuscript. JDL, WSD, and ACL contributed to the research design and writing of the manuscript. CH and LE reviewed and contributed to the manuscript writing.

*Competing interests.* The authors declare no competing interest.

*Acknowledgements.* SKG is supported by the Swiss Federal Office for the Environment (FOEN) and the Natural Environment Research Council (NERC) while holding associate status at the University of York.





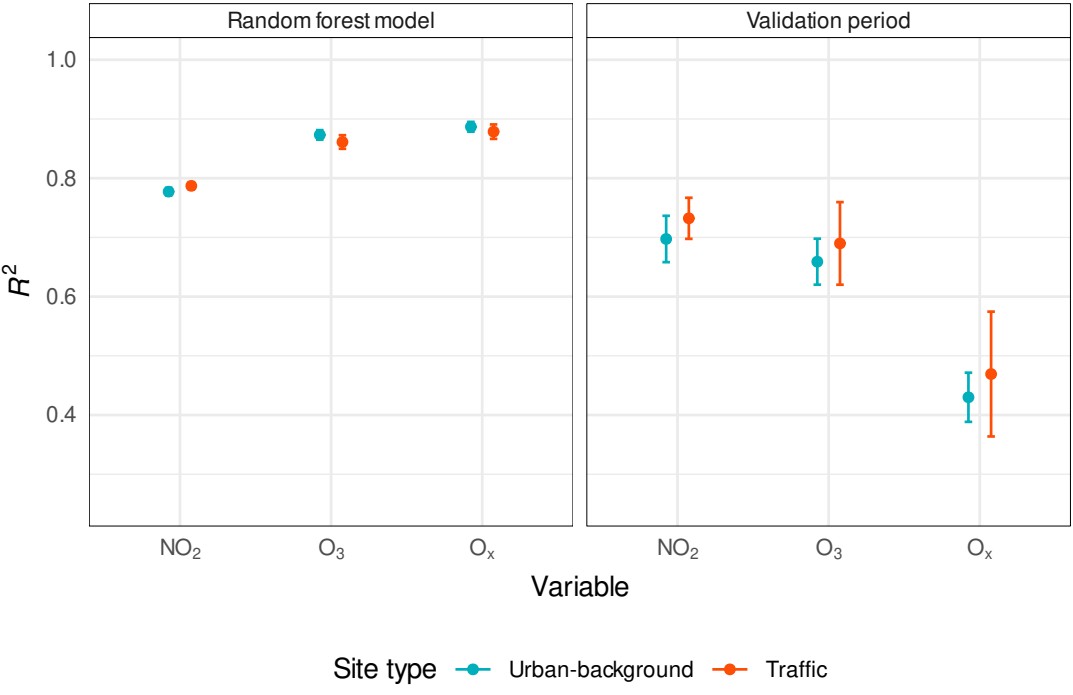

**Figure A1.** Summaries of $R^2$ values from the random forest model objects and for predictions during the model validation period (February 14 to March 1, 2020) for the three predicted variables.



**Figure A2.** Oxford COVID-19 Government Response Tracker's (OxCGRT) stringency index of COVID-19 lockdown measures imposed by different countries' governments between February and July, 2020 (Hale et al., 2020).





**Figure A3.** Volatile organic compounds (VOCs) time series at London Eltham, an urban-background site in United Kingdom.



**Table A1.** Most commonly identified dates where observed and BAU modeled concentrations diverged in March, 2020. Dates which are missing indicates no change point was detected in March, 2020.

| Country | Lockdown date | $NO_2$ date | $O_3$ date |
|---|---|---|---|
| Andorra | Fri., Mar. 13, 2020 | Sat., Mar. 14, 2020 | Thu., Mar. 19, 2020 |
| Austria | Mon., Mar. 16, 2020 | Thu., Mar. 19, 2020 | Mon., Mar. 16, 2020 |
| Belgium | Wed., Mar. 18, 2020 | Sun., Mar. 15, 2020 | Sat., Mar. 21, 2020 |
| Bosnia and Herzegovina | Sat., Mar. 21, 2020 | Thu., Mar. 19, 2020 | Thu., Mar. 12, 2020 |
| Bulgaria | Fri., Mar. 13, 2020 | Wed., Mar. 11, 2020 | Wed., Mar. 18, 2020 |
| Croatia | Thu., Mar. 19, 2020 | Fri., Mar. 20, 2020 | Fri., Mar. 20, 2020 |
| Cyprus | Sun., Mar. 15, 2020 | Fri., Mar. 13, 2020 | Thu., Mar. 19, 2020 |
| Czechia | Mon., Mar. 16, 2020 | - | Fri., Mar. 20, 2020 |
| Denmark | Fri., Mar. 13, 2020 | Fri., Mar. 27, 2020 | Tue., Mar. 17, 2020 |
| Estonia | Fri., Mar. 13, 2020 | Mon., Mar. 16, 2020 | Sat., Mar. 21, 2020 |
| Finland | Mon., Mar. 16, 2020 | Tue., Mar. 17, 2020 | - |
| France | Tue., Mar. 17, 2020 | Sat., Mar. 14, 2020 | Wed., Mar. 11, 2020 |
| Germany | Sun., Mar. 22, 2020 | Sun., Mar. 22, 2020 | Sat., Mar. 28, 2020 |
| Greece | Mon., Mar. 16, 2020 | Tue., Mar. 17, 2020 | - |
| Hungary | Mon., Mar. 16, 2020 | - | Sat., Mar. 14, 2020 |
| Iceland | Mon., Mar. 16, 2020 | Sat., Mar. 14, 2020 | - |
| Ireland | Fri., Mar. 13, 2020 | Thu., Mar. 19, 2020 | - |
| Italy | Mon., Mar. 09, 2020 | Fri., Mar. 13, 2020 | Thu., Mar. 19, 2020 |
| Lithuania | Mon., Mar. 16, 2020 | Tue., Mar. 17, 2020 | Wed., Mar. 11, 2020 |
| Luxembourg | Mon., Mar. 16, 2020 | Sat., Mar. 14, 2020 | Fri., Mar. 20, 2020 |
| Malta | Sun., Mar. 22, 2020 | Sat., Mar. 14, 2020 | Sun., Mar. 15, 2020 |
| Netherlands | Mon., Mar. 16, 2020 | Mon., Mar. 16, 2020 | - |
| North Macedonia | Wed., Mar. 18, 2020 | Fri., Mar. 13, 2020 | - |
| Norway | Thu., Mar. 12, 2020 | Tue., Mar. 17, 2020 | - |
| Poland | Thu., Mar. 12, 2020 | Tue., Mar. 17, 2020 | Tue., Mar. 24, 2020 |
| Portugal | Wed., Mar. 18, 2020 | Sat., Mar. 14, 2020 | Wed., Mar. 18, 2020 |
| Romania | Mon., Mar. 16, 2020 | Sat., Mar. 21, 2020 | Tue., Mar. 17, 2020 |
| Serbia | Sat., Mar. 21, 2020 | Tue., Mar. 17, 2020 | Mon., Mar. 16, 2020 |
| Slovakia | Mon., Mar. 16, 2020 | Sun., Mar. 22, 2020 | - |
| Slovenia | Mon., Mar. 16, 2020 | Thu., Mar. 12, 2020 | Tue., Mar. 17, 2020 |
| Spain | Sat., Mar. 14, 2020 | Sat., Mar. 14, 2020 | Sun., Mar. 15, 2020 |
| Sweden | - | Wed., Mar. 18, 2020 | Fri., Mar. 20, 2020 |
| Switzerland | Tue., Mar. 17, 2020 | Sun., Mar. 22, 2020 | Thu., Mar. 26, 2020 |
| United Kingdom | Mon., Mar. 23, 2020 | Mon., Mar. 23, 2020 | Thu., Mar. 26, 2020 |



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
