# Peer review of "COVID-19 lockdowns highlight a risk of increasing ozone pollution in European urban areas"

_Atmospheric Chemistry and Physics, 2020_

## Referee Comment (RC1) · Anonymous Referee #1 · 18 Dec 2020

This paper analyzed the effect of the European COVID-19 lockdowns on NO2, O3, and Ox concentrations by comparing the observation and business as usual (BAU) derived from machine learning at 246 stations. The lockdown effect was determined by the Bayesian change point models. This analyze gave an 34% reduction of NO2 concentration and 30% increase of O3 leading to little change in Ox. Therefore, the change in NO2 and O3 is mainly a repartitioning of Ox. This paper presents a timely and important analysis of evaluating the lockdown impact on air quality in Europe. The paper is well written and structured. I suggest the authors to consider the following comments, which may help to improve the paper.

General Comments:

1. The description of methods are not detailed enough. Although most of the methods

are used and described in previous studies, more detail information, for example about how the calculation of BAU, is useful for the reader to understand the data processing. There are some references provided to show how to perform the data analysis but are written in a specific programming language. The fundamental description in the paper would be helpful in case some of the readers are not using this tool.

2. The input and output of random forest model is hourly data. This is different from Grange et al. (2018), where daily averaged data was used. Why is the change? If hourly data is used, I am not sure how well the model captures the lockdown effect if the variability is mainly contributed by diurnal variation. Also, it would be very helpful to show the performance of the BAU calculation in a time series plot in their absolute concentration, perhaps in the supplement. The current comparison is only show in Fig A1 with some averaged $R^2$ is not enough. At least the performance for different countries should be show individually unless the model performances are the same. The calculation of Ox BAU is not clear. Is it calculated from the Ox observation like NO2 and O3, or the sum of NO2_BAU and O3_BAU?

3. The argument of ozone pollution need for evidence to support. As discussed in section 3.5, the increase in O3 is mainly a repartitioning of Ox during the lockdown. The Ox/O3/NO2 concentrations were missing so I cannot tell from the paper itself if all Ox are in the form of O3, will O3 exceed the limit? This is a rough estimation assuming only repartitioning play a role. As mentioned in the paper, the ozone formation is nonlinear with VOC and NOx and Europe is likely in the VOC-limited regime. Reduction in NOx do not lead to higher O3 formation. If the reduction in NOx is stronger than lockdown in the future, ozone production could move to NOx-limited regime, which ozone pollution less important.

Technical comments:

Line 109: The model prediction is corrected by -3.7ug for NO2. How much you result sensitive to this correction.

[Figure]

Line 112: The underprediction of NO2 is attributed to mild temperature and windy conditions. Isn't this indicating the model is not able to predict the condition in 2020?

Line 210—213: The projection of NO2 reduction and O3 increase in the future is assuming a linear trend, which seems to me a bit too simple. Especially calculating the year to reach the lockdown impact bothers me.

Line 229—230: This sentence is not clear.

Line 233: maybe you want to refer your argument to table 1.

Line 274: I think better to state the access of both NO and NO2 data.

---

## Referee Comment (RC2) · Shaojun Zhang (Referee) · 30 Dec 2020

Grange et al. utilized time-series random forest models to analyze the changes of NO2 and O2 concentrations caused by the COVID-19 lockdowns across European countries. This work has important findings from the natural experiment of atmospheric pollution that most urban areas in Europe is in the VOC-limited scheme of O3 formation (e.g., at least in Spring). Therefore, only mitigating traffic NOx emissions might bring in unwanted increase of urban O3. Overall, the manuscript is well organized, and the data analysis is solid and consistent.

Line 29: I suggest add the explanation of the evaluation metric of Google mobility; e.g., the search frequency of points of interest, or the visit frequency (or duration spent) at

points of interest?

Line 36: Please reconsider the wording "near-minimum". I suppose commercial, transportation and recreation activities would be drastically declined, and the impact on essential industrial sectors would be less substantial.

Line 57: Please describe the distance between traffic sites and urban-BG sites in the selected urban areas. I wonder whether these traffic sites in various European countries would be deployed based on a unified, clear principle (e.g., distance to road curb, daily traffic volume)? Or, consider to enhance the statement around Line 70.

Line 65: Please briefly describe how to match air quality and weather sites in this study. Line 104: It is not clear, in Figure A1, whether the distribution of R2 represents the interval of R2 (minimum to maximum) for each site-specific RF model? In addition to R2, other validation metrics like normalized mean error can be used to evaluate the average discrepancy between modelled and observed results. And, I am surprised that both NO2 and O3 share good model validation results but Ox has lower R2. What are the possible reasons and implications?

Line 109: what is the percentage of underestimation.

Line 147: What is the possible cause (from the perspective of atmospheric chemistry or model validation performance) of comparable O3 concentrations in the late period of this analysis to the business-as-usual levels, while NO2 concentrations still indicated some degree of NOx emission reduction?

Line 170: I consider the less correlated relationship between lockdown date and O3 surge possible is because O3 is a more regional pollutant than NO2 (high contribution from regional transport). I wonder how about analyzing the maximum daily average 8-hr instead of all O3 observations?

Line 185: Is there any supporting mobility data to verify the actual change of mobility activities in Germany and Switzerland vs. in France and Italy?

Line 205: Please consider to add the increase of maximum daily average 8-hr ozone concentrations.

Line 210: The authors has strong assumptions that the future reduction pace of NO2 would follow that in the past decade, and the O3 increase would greatly relate to the change of traffic emissions. I am not very confident with these assumptions. In particular, O3 pollution is a regional issue, and is relevant to emission controls not only for NOx but also for VOCs (e.g., deeper mitigation of NOx might lead to O3 reduction). Similar concern for the statement in the abstract (e.g., the predicted situation in 2028)

Line 265: and biogenic VOCs emissions.

Figure A3. What are the measurement methods and data reliability of VOC concentrations?

---

## Author Comment (AC1) · 29 Jan 2021

**Author responses to reviewers' comments of acp-2020-1171 (*COVID-19 lockdowns highlight a risk of increasing ozone pollution in European urban areas*)**

January 29, 2021

Stuart K. Grange*, James D. Lee, Will S. Drysdale, Alastair C. Lewis, Christoph Hueglin, Lukas Emmenegger, and David C. Carslaw

*stuart.grange@empa.ch

**Response to reviewers**

**Anonymous Referee #1**

This paper analyzed the effect of the European COVID-19 lockdowns on $NO_2$, $O_3$, and $O_x$ concentrations by comparing the observation and business as usual (BAU) derived from machine learning at 246 stations. The lockdown effect was determined by the Bayesian change point models. This analyze gave an 34% reduction of $NO_2$ concentration and 30% increase of $O_3$ leading to little change in $O_x$. Therefore, the change in $NO_2$ and $O_3$ is mainly a repartitioning of $O_x$. This paper presents a timely and important analysis of evaluating the lockdown impact on air quality in Europe. The paper is well written and structured. I suggest the authors to consider the following comments, which may help to improve the paper.

Thank you for your positive comments and suggestions for improvements. Please see the itemised responses below.

**General comments**

1. The description of methods are not detailed enough. Although most of the methods are used and described in previous studies, more detail information, for example about how the calculation of BAU, is useful for the reader to understand the data processing. There are some references provided to show how to perform the data analysis but are written in a specific programming language. The fundamental description in the paper would be helpful in case some of the readers are not using this tool.

We have expanded the methods section to describe the method's approach further. The previous papers we have references give comprehensive details on the methods used and therefore, here, we focus on explaining the calculation of the business as usual scenario. The new text now reads:

"The philosophy of this approach involves using a machine learning model, trained on past data, to predict beyond the last observations it has seen. The model is trained on a

long enough period, two years in this work, to capture the variability of concentrations experienced in a variety of meteorological conditions. Beyond the training period (February, 14, 2020), the model predicts concentrations based on meteorological variables which from the model's perspective are from the future. The time series which results is a *counterfactual*. This counterfactual represents an estimate of concentrations during a business as usual (BAU) scenario. The BAU concentrations can be readily compared with what was observed for example, Figure 3 and the changes quantified, explained, and interpreted. This allows for a robust comparison with what was expected with what was observed."

2. The input and output of random forest model is hourly data. This is different from Grange et al. (2018), where daily averaged data was used. Why is the change? If hourly data is used, I am not sure how well the model captures the lockdown effect if the variability is mainly contributed by diurnal variation. Also, it would be very helpful to show the performance of the BAU calculation in a time series plot in their absolute concentration, perhaps in the supplement. The current comparison is only show in Fig A1 with some averaged $R^2$ is not enough. At least the performance for different countries should be show individually unless the model performances are the same. The calculation of $O_x$ BAU is not clear. Is it calculated from the $O_x$ observation like $NO_2$ and $O_3$, or the sum of $NO_2\_BAU$ and $O_3\_BAU$?

Hourly data were used because (a), they result in more performant models when compared to lower resolution models, chiefly because explanatory variables such as wind direction have far more information at higher resolution and (b), these hourly data are available for the analysis time period. Grange et al. (2018) analysed $PM_{10}$ data over a much longer period and these data were collected by gravimetric samplers and therefore, were at daily resolution.

We agree that additional $O_3$ metrics (such as rolling 8-hour means) might capture some attributes which maybe somewhat hidden by the mean response. However, we would argue that the mean response of all species is the most important metric and will represent the changes in concentrations well.

Displaying the sites' time series is not practical in a publication such as this. However, we have provided an example time series in concentration units in Figure 3. Figure 3 very clearly demonstrates the counterfactual divergence from the observed concentrations.

We agree that the lack of model error statistics was a weakness in the manuscript. We have addressed this by replacing Figure A1 with a more comprehensive version (also below in Figure 1). The new figure shows Pearson's correlation coefficient ($r$), mean bias

(MB), normalised mean bias (NMB), and normalised root mean square error (NRMSE) for all sites' models for the training and validation periods.

The $O_x$ modelling was indeed done with observational data like $NO_2$ and $O_3$. This statement was missing in the manuscript and has been edited:

"... random forest models were trained to explain hourly mean $NO_2$, $O_3$, and $O_x$ concentrations using surface meteorological and time explanatory variables for each monitoring site."

3. The argument of ozone pollution need for evidence to support. As discussed in section 3.5, the increase in $O_3$ is mainly a repartitioning of $O_x$ during the lockdown. The $O_x/O_3/NO_2$ concentrations were missing so I cannot tell from the paper itself if all $O_x$ are in the form of $O_3$, will $O_3$ exceed the limit? This is a rough estimation assuming only repartitioning play a role. As mentioned in the paper, the ozone formation is nonlinear with VOC and $NO_x$ and Europe is likely in the VOC-limited regime. Reduction in $NO_x$ do not lead to higher $O_3$ formation. If the reduction in $NO_x$ is stronger than lockdown in the future, ozone production could move to $NO_x$-limited regime, which ozone pollution less important.

The reviewer rightly emphasizes the non-linear nature of its production involving $NO_x$ and VOCs. However, the focus of the current study is constrained to urban areas including roadside locations, rather than regional scale rural locations. To understand this issue more fully at a European scale would require air quality modelling. However, as shown in Figure 6, $O_x$ concentrations varied little at background and traffic sites in comparison with either $NO_2$ or $O_3$. As discussed elsewhere in the paper, (and shown Figure 6), there is more evidence of $O_x$ concentrations decreasing at traffic sites, which can be attributed to reductions in the primary emission of $NO_2$.

**Technical comments**

1. Line 109: The model prediction is corrected by -3.7ug for $NO_2$. How much you result sensitive to this correction.

We apologise, but we do not understand this question. However, we believe that what is being asked is "how sensitive are your results to the bias correction applied to $NO_2$". The correction applied resolved the systematic underprediction of $NO_2$ due to already lower emissions before the lockdowns were officially implemented — see response to Line 112 below where the use of a correction is used primarily to support a consistent representation of changes relative to a specific date.

[Figure]

Figure 1: Model error summaries for all monitoring sites' (coded as integers) $NO_2$, $O_3$, and $O_x$ models for two datasets – the training and validation sets. The error summaries are Pearson's correlation coefficient ($r$), mean bias (MB; in $\mu g\, m^{-3}$), normalised mean bias (NMB), and normalised root mean square error (NRMSE). The normalised were normalised by the observed mean.

2. Line 112: The underprediction of $NO_2$ is attributed to mild temperature and windy conditions. Isn't this indicating the model is not able to predict the condition in 2020?

   A substantial component of the under prediction would be due to already curtailed economic activity and therefore, emissions before the introduction of lockdowns, and is primarily used to provide a consistent basis for quantifying the changes relative to a specific point in time. The bias correction mostly resolved these issues and the model performance was very good (see Figure A1). The random forest modelling approach used has limitations on what it can achieve. The under prediction of $NO_2$ 2020 can be partially explained by the rather unusual weather conditions experienced in the spring of 2020 which the model could not represent perfectly.

3. Line 210–213: The projection of $NO_2$ reduction and $O_3$ increase in the future is assuming a linear trend, which seems to me a bit too simple. Especially calculating the year to reach the lockdown impact bothers me.

   The goal of these calculations and discussion was put the changes observed in 2020 into context – they do not have the objective of being predictions. We have added a sentence explicitly stating this:
   "These calculations have not been done to predict future concentrations, only to put the changes experienced between March and July, 2020 in context."
   The abstract has also been slightly edited for clarity around this point.

4. Line 229–230: This sentence is not clear.

   We have edited for this sentence for clarity and it now reads:
   "The roadside increment in $NO_2$ above urban background concentrations diminished considerably over lockdown due to large reductions in vehicle activity."

5. Line 233: maybe you want to refer your argument to table 1.

   Done

6. Line 274: I think better to state the access of both NO and $NO_2$ data.

   Done.

**Referee: Shaojun Zhang Referee #2**

Grange et al. utilized time-series random forest models to analyze the changes of $NO_2$ and $O_3$ concentrations caused by the COVID-19 lockdowns across European countries. This work has important findings from the natural experiment of atmospheric pollution that most urban areas in Europe is in the VOC-limited scheme of $O_3$ formation (e.g., at least in Spring). Therefore, only mitigating traffic $NO_x$ emissions might bring in unwanted increase of urban $O_3$. Overall, the manuscript is well organized, and the data analysis is solid and consistent.

Thank you for your positive comments. Please see the itemised responses below.

1. Line 29: I suggest add the explanation of the evaluation metric of Google mobility; e.g., the search frequency of points of interest, or the visit frequency (or duration spent) at points of interest?

   The Google mobility data is highly anonymised and only reports "movement trends" in contrast to a baseline. The text and the figure caption has been altered to clearly explain this:

   "Google's mobility data (Google, 2020) based on movement trends very effectively demonstrates the change in mobility based on a baseline (Figure 1)."

2. Line 36: Please reconsider the wording "near-minimum". I suppose commercial, transportation and recreation activities would be drastically declined, and the impact on essential industrial sectors would be less substantial.

   The text has been altered and now reads:
   "The European lockdowns can be thought of and approached as an air quality 'experiment' where economic activity was substantially curtailed where commercial, transportation, and recreation activities drastically declined."

3. Line 57: Please describe the distance between traffic sites and urban-BG sites in the selected urban areas. I wonder whether these traffic sites in various European countries would be deployed based on a unified, clear principle (e.g., distance to road curb, daily traffic volume)? Or, consider to enhance the statement around Line 70.

   We have calculated the distances among the different urban-traffic and urban-background sites within each urban area. The mean and median distances among the sites in an urban area was 5.2 and 3.9 km respectively. Therefore, the majority of the sites were in rather close proximity to one another and offer good comparisons to one another. The text has been updated to reflect this:

*"The mean distance among the different air quality monitoring sites within an urban area was 5.2 km."*

*The site-type classification was done on the data contained in the European Air Quality e-Reporting (AQER) database. Although the classifications of the sites may contain differences among the authorities supplying data to AQER database, efforts are made to apply the same classification system among all member states.*

4. Line 65: Please briefly describe how to match air quality and weather sites in this study.

   *The text has been updated to describe the matching logic:*

   *" The matching logic between the air quality and meteorological sites was simple. The nearest ISD site to a particular air quality site was determined, the observations queried, and tested to ensure the data record was complete for the analysis period."*

5. Line 104: It is not clear, in Figure A1, whether the distribution of $R^2$ represents the interval of $R^2$ (minimum to maximum) for each site-specific RF model? In addition to $R^2$, other validation metrics like normalized mean error can be used to evaluate the average discrepancy between modelled and observed results. And, I am surprised that both $NO_2$ and $O_3$ share good model validation results but $O_x$ has lower $R^2$. What are the possible reasons and implications?

   *Figure A1 has been replaced with a more comprehensive version to address the limited information supplied in the original manuscript (see Figure 1 and Referee #1's second general comment). The updated version displays more metrics, including normalised mean bias and normalised root mean square error.*

   *It is indeed interesting to see that the predictive performance of $O_x$ is somewhat lower than $NO_2$ and $O_3$. This can be explained by $O_x$ displaying less of a diurnal cycle than other pollutants. The somewhat invariant concentrations result in the "hour" variable having less information gain when compared to the other modelled variables.*

6. Line 109: what is the percentage of underestimation.

   *The mean percentage change has now been added to the text:*

   *"The under-prediction was on average, -3.7 $\mu g\,m^{-3}$ (95 % CI: [-4.2, -3.3]; mean percentage change: 15.9 %)."*

7. Line 147: What is the possible cause (from the perspective of atmospheric chemistry or model validation performance) of comparable $O_3$ concentrations in the late period of this analysis to the business-as-usual levels, while $NO_2$ concentrations still indicated some degree of $NO_x$ emission reduction?

We believe that actual $NO_2$ and $NO_x$ emissions remained lower than the business-as-usual scenarios across Europe until the end of July, 2020. The difference between the observed and predicted $NO_2$ concentrations at traffic sites was greater than the urban-background sites. This suggests that traffic emissions are mostly responsible for this. Despite the reduction in emissions, there was still adequate $NO_x$ in European urban atmospheres to generate business-as-usual $O_3$ concentrations by the end of the analysis period (July 31, 2020).

8. Line 170: I consider the less correlated relationship between lockdown date and $O_3$ surge possible is because $O_3$ is a more regional pollutant than $NO_2$ (high contribution from regional transport). I wonder how about analyzing the maximum daily average 8-hr instead of all $O_3$ observations?

We also believe that analysing additional $O_3$ metrics may provide some value. However, analysing the mean response is the most useful and provides consistency with the other species included in the analysis. The extension of the analysis to additional $O_3$ metrics would be better handled in future work and studies.

9. Line 185: Is there any supporting mobility data to verify the actual change of mobility activities in Germany and Switzerland vs. in France and Italy?

The stringency indices (Figure A2) indicate that the countries such as France and Italy had a more strict collection of policies when compared to other countries such as Germany and Switzerland. The © Google's mobility also support this (Figure 2).

10. Line 205: Please consider to add the increase of maximum daily average 8-hr ozone concentrations.

Please see our above response to question/point 8.

11. Line 210: The authors has strong assumptions that the future reduction pace of $NO_2$ would follow that in the past decade, and the $O_3$ increase would greatly relate to the change of traffic emissions. I am not very confident with these assumptions. In particular, $O_3$ pollution is a regional issue, and is relevant to emission controls not only for $NO_x$ but also for VOCs (e.g., deeper mitigation of $NO_x$ might lead to $O_3$ reduction). Similar concern for the statement in the abstract (e.g., the predicted situation in 2028).

The same question has been asked by Referee #1 (technical comment 3). The goal of these calculations and discussion was put the changes observed in 2020 into context – they do not have the objective of being predictions. We have added a sentence explicitly stating

[Figure]

Figure 2: © Google's mobility indices between February and July, 2020 for four selected countries split by subjective groups.

this:

"These calculations have not been done to predict future concentrations, only to put the changes experienced between March and July, 2020 in context."

12. Line 265: and biogenic VOCs emissions.

Done.

13. Figure A3. What are the measurement methods and data reliability of VOC concentrations?

The observations displayed in Figure A3 were gained from the Automatic Hydrocarbon Network. This network uses automatic gas chromatographs to report real-time speciated hydrocarbons. These data are reported to the European Commission and are considered high-quality. For more information, please visit `https://uk-air.defra.gov.uk/networks/network-info?view=hc`.

**Other changes**

- The data file which was provided as a temporary link has been migrated to a persistent data repository (https://doi.org/10.5281/zenodo.4464734). This repository is now referenced in text and in the data availability section.

- Three additional references have been added which report air quality changes due to COVID-19 lockdown measures.

- Table 1 had a formatting error which resulted in duplicate rows. This has been fixed.

[revised manuscript text omitted]